# Facile Synthesis of Natural Anise-Based Nanoemulsions and Their Antimicrobial Activity

**DOI:** 10.3390/polym13122009

**Published:** 2021-06-19

**Authors:** Ola A. Abu Ali, Mehrez E. El-Naggar, Mohamed S. Abdel-Aziz, Dalia I. Saleh, Mohamed. A. Abu-Saied, Wael A. El-Sayed

**Affiliations:** 1Department of Chemistry, College of Science, Taif University, P.O. Box 11099, Taif 21944, Saudi Arabia; o.abuali@tu.edu.sa (O.A.A.A.); daliarawan@yahoo.com (D.I.S.); 2Textile Research Division, National Research Center, Dokki, Giza 12622, Egypt; 3Genetic Engineering and Biotechnology Division, National Research Centre, Dokki, Giza 12622, Egypt; abdelaziz146@gmail.com; 4Polymeric Materials Research Department, Advanced Technology and New Materials Research Institute, City of Scientific Research and Technological Applications (SRTA-CITY), New Borg El-Arab City 21934, Alexandria, Egypt; mouhamedabdelrehem@yahoo.com; 5Department of Chemistry, College of Science, Qassim University, Buraidah 52571, Saudi Arabia; waelshendy@gmail.com; 6Photochemistry Department, National Research Centre, Dokki, Cairo 12622, Egypt

**Keywords:** anise oil, nanoemulsion, emulsifying agent, antimicrobial properties

## Abstract

Anise oil was prepared in its nanoemulsion form to facilitate the penetration of microbial walls, causing microbe mortality. The penetration occurred easily owing to the reduction in its size (nm). Nanoemulsions with different concentrations of anise oil were prepared using lecithin as an emulsifying agent with the aid of an ultra-sonification process. Their morphological and chemical properties were then characterized. The promising constituents were l-Menthone (11.22%), Gurjunene (6.78%), Geranyl acetate (4.03%), Elemene (3.93%), Geranyl tiglate (3.53%), geraniol (3.48%), linalool (0.17%) as well as camphene (0.12%). Different concentrations of prepared anise oil in micro and nanoemulsions were tested as antimicrobial agents against Gram-positive bacteria (*Staphylococcus aureus*), Gram-negative bacteria (*Escherichia coli*), yeast (*Candida albicans*) and fungi (*Asperigillus niger*). The findings illustrated that the anise oil-based nanoemulsion exhibited better results. Different biochemical and biological evaluations of anise oil nanoemulsions were conducted, including determining killing times, antioxidant activities (using three different methods), and total phenolics. A trial to estimate the mode of action of anise oil-based nanoemulsion as an antimicrobial agent against *S. aureus* and *C. albicans* was performed via studying the release of reducing sugars and protein and conducting scanning electron microscopy.

## 1. Introduction

Essential oils are natural products that are used in the preservation of foods [1]. The Food and Drug Administration (FDA) classifies essential oils as safe for food consumption [2,3]. Therefore, the antimicrobial activity of plant extracts therefore relies on the existence of phenolic compounds in essential oils [4]. For decades, oil-in-water emulsions have been recognized as efficient systems for hydrophobic compounds by dispersing the lipid process as a colloidal dispersion [5]. As a natural aromatic herb and spice, anise has been in use since ancient times, and its seeds are used as an ingredient in folk medicine and fruit. Chemical studies have shown that anise contains estragole, anethole, scopoletin, eugenol, coumarin, pseudoisoeugenol, anisaldehyde, methyl chavicol, umbelliferone, scopoletin, terpene, estrol, polyacetylene, and pole [6].

As one of the most significant technological developments, nanotechnology has been widely applied in many fields, including medicine, agriculture, textiles, and the food industry, because of its considerable potential to develop antimicrobial delivery systems [4,7]. Nanoemulsions are defined as metastable dispersions of submicron oil in water with a droplet diameter within the 10–100 nm range [8]. The potential benefit of nanoemulsions as carriers versus traditional delivery systems is the improvement of biocompatibility and penetration to the desired active site [9,10] due to the high surface area and the small size of the formed nanodroplets along with high stability during storage. The well stability of these nanodroplets makes the whole nanoemulsion suitable for use in many different domains, unlike the poor stability and the large size of the materials in their microemulsion state.

The pharmacopeia requirements for drug delivery of nanoemulsions consist of the following items: sterility, isotonicity, nonpyrogenic compounds, nontoxic compounds, biodegradability, and also stability. Consequently, in comparison with traditional emulsions, nanoemulsions have a much more desirable toxicological profile. They also have certain special features, having a high colloidal stability and being non-irritant and nontoxic. All these specific properties must be taken into consideration when targeting the nanoemulsions as a drug delivery system. Hence, for clinical purposes, numerous nanoemulsions are currently viable for drug delivery domains. Nanoscale droplets are being used as a drug reservoir for sustained release and can also strengthen the pharmacokinetics and targeting of drugs [11,12,13]. In addition, nanoemulsions have high loading for lipophilic and hydrophilic drugs. Therefore, it is important to design effective delivery systems by incorporating essential oils and studying how the different formulations may affect their antimicrobial activities.

Ultrasonic emulsification is an energy-intensive method of nanoemulsion development, which is a fast and efficient technique for formulating stable nanoemulsions with a small diameter size and low polydispersity [14,15]. Therefore, when designing effective delivery systems, it is necessary to incorporate essential oils and research how the different formulations can influence their antimicrobial activities. Several essential oils were investigated for their antimicrobial activities [16]. Essential oils from *Mentha piperita, Lavandula angustifolia, Mentha pulegium, Salvia lavandulifolia,* and *Satureja montana* were studied for their biological activities, including antimicrobial activities and cytotoxicity [17]. It has been reported that the essential oils that originated from *Organum vulgare* were tested against *E. coli* and the obtained data illustrated that the minimum inhibition concentration (MIC) value recorded 1600–1800 pp. On the other hand, essential oil from lavender had a MIC value of 1000–1200 pp when tested against *S*. *aureus* [18].

This study aimed to explore the influence of anise oil concentration and emulsifier on physical properties such as particle shape, morphology, hydrodynamic particle size, polydispersity index (PDI), and zeta potential values of the prepared anise oil-based nanoemulsion in order to obtain significant information about the droplet diameter. In addition to investigating the activity of bulk anise oil against many types of pathogenic microbes, the research work was expanded to test antimicrobial nanoemulsion-based activities, which were also tested when prepared at different concentrations.

## 2. Materials and Methods

### 2.1. Materials

Anise oil with a high purity (>99%) was locally obtained from our traditional remarkable market (Giza, Egypt). Meanwhile, lecithin was purchased from Fischer Co. (Dallas, TX, USA). Milli-Q (Millipore Corporation, Burlington, MA, USA) water was used for synthesis in all experiments, characterization, and antimicrobial evaluations.

### 2.2. Preparation of Anise Oil-Based Nanoemulsion

Anise oil-based nanoemulsion was synthesized by means of using lecithin as an emulsifying agent, as follows: lecithin (7%) was dissolved in chloroform at room temperature under magnetic stirring for 15 min. After lecithin dissolution, anise oil with different concentrations (5.5, 9.5, and 13.5 mL) was added dropwise to 100 mL of lecithin using chloroform as a solvent to obtain different ratios of chloroform:lecithin:anise-oil (1:0.07:0.055, 1:0.07:0.095, and 1:0.07:0.13, respectively). The fabricated emulsions were kept under homogenization process using a high-speed homogenizer (Stansted, Essex, UK, operating from 22 to 88 Mpa) for 15 min at room temperature. After that, the homogenized mixture was subjected to an ultra-sonication process (80 kHz) and output powers (200 W) for 30 min at room temperature. Ultimately, the three resulting cloudy white color anise oil-based nanoemulsions were processed at room temperature under dark conditions in a laboratory incubator. The cloudy white color of the nanoemulsion was owing to the spontaneous formation of tiny oil droplets.

### 2.3. Characterization of Bulk Anise Oil and Anise Oil-Based Nanoemulsion

#### 2.3.1. Chemical Analysis of Anise Oil Using Gas Chromatography–Mass Spectrometry (GS-MS) Spectroscopy

GC-MS analysis was conducted using an Agilent 6890 N GC/5975MSDSCAN (Agilent Technologies, Palo Alto, CA, USA) as described by Adams [19] and Ghareeb et al. [20]. The produced mass spectra of anise oil were examined by comparing them to the mass spectra of 392,086 compounds from the Wiley 7 N library and 174,948 compounds from the NIST 2002 library.

#### 2.3.2. Physical Characterization of Anise-Based Nanoemulsion

Transmission electron microscopy (TEM; JEOL, JEM 2100, 120 KV acceleration voltage, Peabody, MA, USA) was utilized to assess the morphology and structure of the anise-based nanoemulsion. Drops of the nanoemulsion samples of anise was diluted with deionized water 20 times. On a 400-mesh formvar carbon film-coated copper grid, a droplet of diluted nanoemulsion was mounted. Then, uranyl acetate stained the grid adversely. To remove the excess liquid, the grid was placed on a sheet of Whatman filter paper. The grid was then ready for analysis by TEM.

The average hydrodynamic size, polydispersity, and zeta potential of the prepared nanoemulsion were examined by means of dynamic light scattering (DLS, Zetasizer Nano-ZS, Malvern Instruments, London, UK). The samples of the synthesized nanoemulsions were diluted using deionized water at 1:100. All these measurements for particle size distribution and zeta potentials were conducted three times at 25 °C.

The morphological features of the anise-based nanoemulsions were examined via field emission-scanning electron microscope (FE-SEM QUANTA FEG250, Czech, Republic). The anise oil-based nanoemulsions were retained and vortexed for approximately 5 min. On metal stubs, a suspension drop was placed using double-sided adhesive tape, left to dry, and then coated in vacuum with Au (350 Å) in a Spi-module Sputter Coater; EDEN Instruments, (Bordeaux, France), followed by a straightforward examination by FE-SEM.

#### 2.3.3. Antimicrobial Activity by Cup Plate Agar Diffusion Method For Macro and Nanoemulsion from Anise Oil

Cup plate diffusion technique was used to assess the antimicrobial actions of anise oil -based nanoemulsion by means of using the tested microbes; Gram-positive bacterium: *Staphylococcus aureus* (*S. aureus*, ATCC 6538), Gram-negative bacterium: *Escherichia coli* (*E. coli*, ATCC 25922), yeast: *Candida albicans* (*C. albicans*, ATCC 10231) and the fungus: *Aspergillus niger* (*A. niger*, NRRL A-326). Both bacteria and yeast were grown on nutrient agar medium, whereas the fungus was grown on Czapek–Dox medium [21,22]. Neomycin and cycloheximide were used as antibacterial and antifungal references, respectively.

#### 2.3.4. Determination of Minimum Inhibition Concentration (MIC) and Minimum Bactericidal Concentration (MBC) of Different Prepared Nanoemulsions

Mueller Hinton (MH) culture broth medium was used for the cultivation of the tested microbes. Resazurin solution was prepared by dissolving 67.5 μg in 10 mL of distilled water (previously sterilized). In sterile 96-well labeled microplates 50 µL of test material was transferred to the first row of the plate. To all wells, 50 µL of MH broth was incorporated. Serial dilutions were performed and 10 µL of resazurin solution was added. Finally, bacterial suspension (10 µL) was added to each well to get concentration of 5 × 10^6^ cfu/mL [23].

#### 2.3.5. Kill-Time

Kill-time analysis was used to examine the bactericidal properties of the nanoemulsions of anise essential oils as mentioned by Joray et al., [24]. Cultivation of test microbes with the anise oil nanoemulsions was performed using the MIC concentration and one of the utilized concentrations mentioned above. Controls having the test microbes only were concurrently run. At certain time intervals, samples were taken, serially diluted in sterile water, and plated in the nutrient agar medium. The inoculated plates were then incubated for 24 h at 37 °C, and colony-forming units (CFUs) were counted.

#### 2.3.6. Antioxidant Activity of Different Concentrations of Anise Nanoemulsion 

2,2′-Azinobis-3-ethylbenzothiazoline-6-sulfonic acid cation radical-scavenging capacity (ABTS). The radical-scavenging capacities of the different concentrations of anise oil nanoemulsions were assayed with ABTS (Sigma Aldrich, Toluca, Mexico) as previously described by Rufino et al. [25]. 2,2-Diphenyl-1-picrylhydrazyl free radical-scavenging capacity (DPPH): Measurement of the DPPH (Sigma Aldrich, Toluca, Mexico) radical-scavenging capacity was performed as described by Karamać et al. [26].

Total antioxidant activity by phosphomolybdenum method: The total antioxidant capacity (TAC) of the anise oil nanoemulsion was investigated as previously carried out by Prieto et al. [27]. A total of 100 μL of anise oil sample solution was mixed with 900 μL of reagent mixture solution (0.6 M sulfuric acid, 28 mM sodium phosphate, and 4 mM ammonium molybdate) (Sigma Aldrich, Toluca, Mexico). For the blank, 100 μL of deionized water was used instead of the nanoemulsion sample. The tubes were kept at 95 °C for 90 min. After cooling, the absorbance of each sample was detected at 695 nm on the spectrophotometer. A standard curve (0.2–1 μg/mL) of ascorbic acid was built to determine the total antioxidant equivalent. The total antioxidant activity was calculated from the following equation:Y = 3.1145x − 0.2523(1)

#### 2.3.7. Determination of the Total Phenolic Contents of Anise-Based Nanoemulsion

Determination of the total phenolic contents gave us an indication of the phenolic contents that were mainly responsible for bioactivity. The total phenolic content was estimated in different concentrations of anise oil nanoemulsion using Folin-Ciocalteu reagent (FCR)-based assay [23]. To 50 μL of plant extract, 950 μL of water, 500 μL of FCR, and 2.5 mL of sodium carbonate solution (20%) were added. The mixture was then kept for 40 min at room temperature and absorbance was estimated at 725 nm. A standard curve of gallic acid was prepared. A control solution was prepared using distilled water instead of extract and the absorbance was recorded against that control. Total phenolic contents (mg/g) of anise oil-based nanoemulsion were expressed as gallic acid equivalent (GAE). Total phenols were calculated from the gallic acid standard curve according to the following equation:y = 0.0116x + 0.074(2)

#### 2.3.8. Cytotoxicity Assay

Cell line: Hepatocellular carcinoma (HEPG-2) and mammary gland breast cancer (MCF-7) were used in the test study. The cell lines were obtained from ATCC via a holding company for biological products and vaccines (VACSERA, Cairo, Egypt). Doxorubicin was used as a standard anticancer drug for comparison.

MTT assay: The cell lines were used to evaluate the inhibitory effects of anise oil nanoemulsions on cell growth using the MTT assay as described by Mosmann [28] and Denizot and Lang [29].

#### 2.3.9. Evaluation of the Mechanism of Action of the Anise Oil Nanoemulsion as Antimicrobial Agent

Release of bacterial cell sugars and proteins: To evaluate the discharge of reducing sugars as well as proteins through bacterial cell membranes, nutrient broth and essential oil nanoemulsions of anise were added to 10 mL of cultures with final concentrations of 125 μg/mL essential oil nanoemulsions and 20 µL of 10^6^ cfu/mL of both *S. aureus* and *C. albicans* [22].

#### 2.3.10. Morphological Structure of Microbial Cell After Subjecting to Anise Oil-Based Nanoemulsion

Anise-based nanoemulsion-treated cells (*S. aureus* and *C. albicans*) were fixed with glutaraldehyde. Cells were washed three times with 0.1 M of PBS as buffer at pH 7.4. Each washing step consumed 15 min and was centrifuged at 10,000 rpm. Amounts of 2.5% (*v*/*v*) glutaraldehyde were mixed with the cells overnight at 4 °C. Cells were washed once more and post-fixed with 1% (*w*/*v*) osmic acid for 2 h at ambient temperature. Cells were washed another time with PBS. Dehydration of cells by consecutive concentrations of ethanol (30, 50, 70, and 90%) was performed followed by acetone (90 and 100%) for 15 min each. After dehydration, inserting medium (Araldite): glycerol-based aromatic resin, which has very little volume shrinkage on polymerization, was added into all samples. The bacteria that were stained with osmic acid were observed and snapped with SEM (FE-SEM, QUANTA FEG250, Czech, Czech Republic) [30]

### 2.4. Statistical Analysis

Analysis of variance (ANOVA) of the data was done by CoStat version 6.4 (CoHort Software, Birmingham, UK). Duncan’s new multiple range test at probability (*p*) level ≤ 0.05 was set to compare among means [31]. Each experiment was carried out three times.

## 3. Results and Discussion

Our research aimed to evaluate, in detail, the antimicrobial properties of anise oil-based nanoemulsion. The antimicrobial properties of anise as bulk molecules have been attributed to the existence of anetholes, which are the primary active component in anise, comprising 97% of its components. Nanotechnology has been developing rapidly for use in all life fields as one of the most significant technological advances of our time [32,33,34,35,36,37,38].

In our work, anise oil was used as a precursor for the formation of anise oil-based nanoemulsion. Lecithin was used as a surfactant for the preparation of the nanoemulsion with the aim of coating and preserving the formed nanoemulsion from further agglomeration and aggregation. The high-speed homogenizer was also utilized to act as a dispersing tool for keeping the forming particles away from each other and keeping them at nanometer size. Additionally, sonication was another tool used for dispersing and preventing the collapse of the produced nanodroplets into large particles [39,40]. Phase separation is the first indication of the formation of a poor nanoemulsion, in addition to storing for a long time. It is well known that the effective preparation of a nanoemulsion is often affirmed by finding no phase separation [14,41]. Based on that, this research was designed to (1) evaluate the effect of different anise oil levels on the physical properties of nanoemulsion (particle size and polydispersity) as a function of storage time, and (2) obtain valuable information about the utilization of such a nanoemulsion in various domains, compare the antimicrobial activities of bulk and nano-emulsified anise oil for common pathogenic bacterial strains, especially food preservation. Below are the characterizations used for clarifying and outlining the particle shape, morphological structure, particle size analyzer, and zeta potential evaluations.

### 3.1. Gas Chromatography–Mass Spectroscopy of Anise Oil

As is well known, the chemical analysis of anise gives an idea about its constituents, which explain its bioactivity. The results in Figure 1 and Table 1 represent the GC-MS of anise oil. The GC-MS analysis identified 51 components representing 99.52% of the total composition. The major components were l-Menthone (11.22%), 1,6-Heptadiene,3,3-dimethyl- (8.35%), 3(R)-Hydroxy-4-acorene (6.79%), Gurjunene (6.78%), 1H-Inden-1-one,octahydro-,cis (6.41%), 2-Pyrrolidinone,1-(4-amino-3,5-dimethylphenyl) (6.29%), Cyclopentanecarboxylic acid, 1-(4-methylphenyl)- (5.19%), 1-Propanol,2-methyl-1-[1-(hydroxymethyl)cyclopropyl] (4.77%), 2-Nonyn-1-ol (4.19%), Geranyl acetate (4.03%), Elemene (3.93%), Geranyl tiglate (3.53%), gamma-Eudesmol (2.97%), and Benzoic acid,5-methylene-1,1,4a-trimethyldecalin-6-on-2-ylester (2.71%).

### 3.2. Characterization of Nanoemulsion Using Transmission Electron Microscopy

TEM technique was conducted at different low and high magnifications in order to outline the effect of lecithin as an emulsifying agent and stabilizing agent for the formed nanodroplets against aggregation. Figure 2A,B displays the TEM of the anise oil-based nanoemulsion prepared using 5.5 mL of pure anise oil in the presence of lecithin. It is evident that the emulsifying agent lecithin acted as a coating agent for the formed anise nanodroplets and acted as a shell for the oil that appeared as a core. Thus, the produced anise oil-based nanoemulsion was well stabilized and protected from agglomeration. Further increasing the concentration of anise oil to 9.5 mL (Figure 2C–E), the potential power of lecithin to act as a stabilizing agent was still good and capable of protecting these nanodroplets from agglomeration with the aid of a high-speed homogenizer. Meanwhile, increasing the concentration of anise oil to 13.5 mL, the produced nanoemulsion (Figure 2F–H) had a different shape. As evident in Figure 2F–H, the nanoemulsion had a spherical shape but there was no evidence of the existence of either core or shell. Such an observation suggests that, at a high concentration of anise oil, the effect of lecithin as an emulsifying agent becomes low, leading to a significant increase in the mean diameters of the nanoemulsion. However, the particles of anise oil-based nanoemulsion were small. From the obtained TEM data, it was clearly noted that the particle sizes formed using 5.5 and 9.5 mL of pure anise oil in the presence of lecithin were smaller than that formed using 13.5 mL of anise oil.

### 3.3. Average Hydrodynamic Particle Size and Zeta Potentials of the Formed Anise-Based Nanoemulsion

The average particle size and zeta potential were utilized to determine the average hydrodynamic size and stability of anise oil-based nanoemulsions obtained using different concentrations of anise oil (5.5, 9.5, and 13.5 mL). The total volume of the emulsion was 100 mL containing 7% lecithin as an emulsifying agent, and the produced anise emulsion coated via lecithin was subjected to high-speed homogenization to further stabilize the formed nanoemulsion. Figure 3A shows that the average hydrodynamic size was small (<60 nm), affirming the positive physical stabilizing effect of lecithin to completely coat the particle of the anise oil-based nanoemulsion and protect it from collapsing into large particles. Moving to the zeta potential graph, as presented in Figure 3B, it was evident that the anise oil-based nanoemulsion had a high value of −35 mv, affirming that the particles of anise oil-based nanoemulsion were stable, even after being stored for a long time (one month), which was further confirmed by the visible eye. It was found that anise oil-based nanoemulsion could stand for one month without noticeable precipitation using 2.5 mL of anise oil as precursor.

The droplet size, as determined by the particle size analyzer, of anise oil-based nanoemulsion prepared using 5.5 mL of pure anise oil (Figure 3C) was very close (68 nm) to the size obtained from Figure 3B, suggesting the formation of a nanoemulsion with a small size. Thus, the droplet size was still significantly unchanged, affirming the substantial power of lecithin to stabilize the produced nanoemulsion, even at a high concentration of 9.5 mL. The nanoemulsion produced with 13.5 mL of anise oil (Figure 3E) had a significantly high particle size of 110 nm.

The polydispersity index (PDI) displays the distribution of the droplets’ particle size. An aim of this study was to determine the conditions required to prepare a nanoemulsion with monodispersity (PDI < 0.5). A nanoemulsion displays polydispersity, meaning that it contains a variety of different particle sizes, when the PDI is more than 0.5 [39,40,41].

Thus, a small PDI value indicates a narrow particle size distribution. The PDI values (0.03, 0.05, and 0.12) displayed in Figure 3A,C,E for the anise oil-based nanoemulsion obtained utilizing 5.5, 9.5, and 13.5 mL of anise oil indicated that all the prepared nanoemulsions had a relatively narrow range of size distribution. The slight increase in the PDI from 0.03 to 0.05 was due to the relative change in the dispersity of the prepared nanoemulsion. An increase in the PDI to 0.2 led to the growth of the oil droplet size of the nanoemulsion with a change in the dispersity of the particles, which, in turn, caused an increase in the particle diameter (Figure 3E).

Characterizing nanoemulsions using their zeta potential is an important tool to confirm the stability of the nanoemulsion. However, the prepared anise oil-based nanoemulsion showed no phase separation, even after one month of synthesis. The zeta potential for the anise oil-based nanoemulsion prepared using 5.5 mL of anise oil was −35 mv (Figure 3B), which was slightly different from the value of −33 mv when the concentration of the utilized anise oil was 9.5 mL, as shown in Figure 3D. The nanoemulsion produced with 13.5 mL of anise oil had a marginally low zeta potential (−31 mv), as presented in Figure 3F.

Overall, the particle size of the prepared anise oil-based nanoemulsion fell within the range of nanometers, and it is expected that its potential application as an antimicrobial agent was not considerably changed. By and large, the value obtained from the TEM, particle size analyzer, and zeta potential confirmed that lecithin has a suitable stabilization affect and could completely cover the surface of oil droplets, resulting in enhanced physical stability.

### 3.4. Morphological Structure of Anise-Based Nanoemulsion

The FE-SEM images used to examine the morphological surface of the anise-based nanoemulsions are displayed in Figure 4. Figure 4A,B illustrates the FE-SEM images of the anise oil-based nanoemulsion that was prepared using 5.5 mL of pure anise oil. It was evident that the prepared nanoemulsion exhibited a spherical shape with a small size. The same observation was made for the nanoemulsion prepared using 9.5 (Figure 4C,D) and 13.5 mL (Figure 4E,F) of pure anise oil. The produced spherical shape indicates that lecithin is an effective emulsifying agent with the help of high-speed homogenization and ultrasonication.

### 3.5. Determination of Antimicrobial Activity for Different Concentrations of Anise-Based Micro- and Nanoemulsion

Previous research has been carried out to outline the effect of anise oil as an antimicrobial and an antioxidant agent [42,43]. Thus, our current study was focused on the biological evaluation of the synthesized anise oil-based nanoemulsion using lecithin as a stabilizing agent. Six concentrations of anise oil-based nanoemulsions were selected according to their antimicrobial activities. These six concentration were coded as 1, 2, 3, 4, 5 and 6. These code samples and their abbreviations are outlined in Table 2. These concentrations were prepared with dilutions of anise oil-based nanoemulsion using 9.5 mL of anise oil.

Thus, these samples were tested for their antimicrobial activities against four different test microbes, namely, *S. aureus*, *E. coli*, *C. albicans,* and *A. niger* belonging to Gram-positive bacteria, Gram-negative bacteria, yeast and fungi, respectively. Results in Table 2 and Figure 5 represent the antimicrobial activity of different nanoemulsions. It was found that all tested nanoemulsions did not show any antimicrobial activity against the Gram-negative bacterium (*E. coli)*. Samples 1, 3, and 4 showed highest antimicrobial activities against *S. aureus,* with inhibition zones of 28, 27, and 24 mm, respectively; *C. albicans* with inhibition zones of 18, 19, and 20 mm, respectively; and *A. niger* with inhibition zones of 22, 24, and 14 mm, respectively. Moderate antimicrobial activities were noticed for sample 2 with an inhibition zone of 22 mm for *S. aureus,* 13 mm for *C. albicans,* and 20 mm for *A. niger.* Samples 5 and 6 showed lower and varied antimicrobial activities with inhibition zones of 12 and 13 mm, respectively, for *S. aureus,* and inhibition zones of 0 and 15 mm, respectively, for *C. albicans,* and antifungal activity with inhibition zones of 0 and 20, respectively, for *A. niger*.

Results represented in Table 3 documented the minimum inhibition concentration (MIC) and minimum bactericidal concentration (MBC) for the as-prepared anise oil-based nanoemulsions. It was found that an anise oil-based nanoemulsion concentration of 0.048 μg/mL exhibited approximately the lowest MIC and MBC values against all test microbes: *S. aureus* (3 and 12 µg/mL), *E.coli* (12 and 24 µg/mL), *C. albicans* (12 and 24 µg/mL), and *A. niger* (24 and 24 µg/mL). Whereas the highest MIC and MBC values were observed with nanoemulsions of 0.165 μg/mL anise oil against *S. aureus* (82.5 and 20.6 µg/mL), *E. coli* (41.25 and 41.25 µg/mL), *C. albicans* (27.1 and 57.5 µg/mL), and *A. niger* (20 and 41.25 µg/mL). Meanwhile, the other concentrations of anise oil-based nanoemulsion exhibited moderate MIC and MBC values compared to the two previously mentioned concentrations (0.048 and 0.165 μg/ mL).

### 3.6. Kill-Time Assay

The killing time of different concentrations of anise-based nanoemulsions was studied. This experiment was carried out for *S. aureus* and *C. albicans*. It was noticed that the number of colonies of *C. albicans* were reduced more than 95% with all concentrations of anise oil-based nanoemulsion after 2 h. However, it was observed that after 24 h, more reductions in growth were found, reaching about 99–100% (Table 4 and Table 5, Figure 6A,B). Moreover, anise oil-based nanoemulsion showed a higher effect on *S. aureus* cells with a reduction percent of about 99% after 2 h, reaching to 100% after 24 h.

The killing time of the tested microbes without treatment with anise oil-based nanoemulsion is presented in Figure 6C. It is clearly seen that the colonies that appeared in plates were uncountable, which affirmed the biological effect of nanoemulsion as an effective matrix for killing pathogenic microbes.

### 3.7. Antioxidant Activity of Different Concentrations of Anise-Based Nanoemulsions Using Different Methods and Evaluations of Their Total Phenolic Contents

The antioxidant of the as-synthesized anise oil-based nanoemulsions were tested using DPPH test. The obtained results showed that the samples, namely, 1, 2, 3, 4, 5, and 6, exhibited antioxidant activities of 13.59, 17.67, and 20.42%, compared with the control as outlined in Figure 7A, and it was observed that these obtained results were slightly higher than those for samples 4 to 6. Therefore, samples 4, 5, and 6 displayed antioxidant activity of 9.89, 6.27, and 5.41%, respectively.

The antioxidant activity of different concentrations of anise oil-based nanoemulsion were also investigated using the 2,2′-Azinobis-3-ethylbenzothiazoline-6-sulfonic acid cation radical-scavenging capacity (ABTS) method. Results in Figure 7B show that samples 1, 2, 3, and 6 had approximately higher antioxidant activity than the other samples, with antioxidant values of 8.51, 10.71, 16.49, and 8.51%, respectively. Lower antioxidant activities were found with samples 4 (5.79%) and 5 (4.67%).

The total antioxidant activity values of different concentrations of anise oil-based nanoemulsion were also investigated. Results in Figure 7C show that samples 2 and 3 had approximately higher antioxidant activity in comparison to the other samples, with antioxidant values of 1.18 and 1.15%, respectively. Lower antioxidant activities were found with samples 1, 4, 5, and 6, with total antioxidant activity 1 (0.84%), 4 (0.74%), 5 (0.87%) and 6 (0.55%).

The total phenolic contents of anise oil-based nanoemulsions were investigated using Folin-Ciocalteu reagent, and gallic acid was used as a control. Additionally, a standard curve was constructed from the findings. The results in Figure 7D represent the total phenolic contents of different nanoemulsions. Sample 3 exhibited the highest total phenolic content of all tested nanoemulsions, with a value of 815.57 µg of GAE/mL, followed by samples 1 (358.6 µg of GAE/mL), 2 (348.2 µg of GAE/mL), 4 (386.37 µg of GAE/mL), 5 (274.2 µg of GAE/mL), and 6 (243.37 µg of GAE/mL). A lower total phenolic content of the tested nanoemulsions was found for samples 5 and 6.

### 3.8. Cytotoxicity Activity of Different Concentrations of Anise Oil-Based Nanoemulsions Against Two Cell Lines, HepG2 and MCF7

The results in Figure 8 and Table 6 show that the anise oil-based nanoemulsion with 55 µg of anise exhibited the lowest IC50 for both cell lines, with values of 9.61 and 7.78 µg for HepG2 and MCF7, respectively. When the original concentration was increased, the IC50 increased for both tested cell lines. The highest IC50 was found for the original concentration of 165 µg, with values of 125.6 and 127.80 µg for HepG2 and MCF7, respectively.

#### 3.8.1. Estimation of the Mode of Action of Different Concentrations of Anise Oil-Based Nanoemulsion by Measuring the Release of Reducing Sugars and Proteins from *S. aureus* and *C. albicans* Treated with Anise Nanodroplets

The results in Figure 9A,B show the release of reducing sugars from *S. aureus* and *C. albicans* cultures treated with anise oil-based nanoemulsion. The released reducing sugars increased from sample 1 to sample 3, with values of 52.785, 78.225, and 98.515 µg/mL, respectively. Samples 4, 5, and 6 exhibited a gradual decrease in reducing sugars with values of 64.515, 59.37, and 57.800 µg/mL, respectively (Figure 9A). However, Figure 9B showed the release of reducing sugars from *C. albicans* cultures treated with anise oil-based nanoemulsions. The released reducing sugars increased from sample 1 to 3 with values of 64.085, 93.925, and 166.23 µg/mL, respectively, whereas samples 4, 5, and 6 exhibited released reducing sugars of 70.66, 169.515, and 103.225 µg/mL, respectively.

The total protein contents that were released from the treated *S. aureus* and *C. albicans* with anise oil-based nanoemulsion were investigated using the Bradford method. The results in Figure 9C show the total protein contents released from *S. aureus* treated with different concentrations of anise oil-based nanoemulsion. Samples 2 and 3 exhibited the highest total protein content of all tested anise oil-based nanoemulsions with values of 45.15 and 32.81 µg/mL, respectively. Samples 4, 5, and 6 exhibited the lowest total protein contents with values of 8.26, 8.26, and 8.02 µg/mL, respectively. The results in Figure 9D show the total protein contents released from *C. albicans* treated with different concentrations of anise oil-based nanoemulsion. Samples 1 and 3 showed the highest total protein content of all tested anise oil-based nanoemulsions with values of 69.195 and 71.23 µg/mL, respectively. Samples 5 and 6 displayed moderate total protein contents with values of 48.26 and 46.17µg/mL, respectively. Meanwhile, samples 2 and 4 showed the lowest total protein contents with values of 3135 and 34.31 µg/mL, respectively.

#### 3.8.2. Effect of Anise Oil-Based Nanoemulsion on the Morphological Structure of Microbial Cells

Scanning electron microscopic studies were performed for both *C. albicans* and *S. aureus* cells before and after treatment with anise oil-based nanoemulsion to ascertain the action of the anise oil-based nanoemulsion as an antimicrobial agent (Figure 10). The results in Figure 10A clarify the morphology of *C. albicans* untreated cells. The cells were intact and had non-scratched cell walls, whereas Figure 10B demonstrates that the anise oil-based nanoemulsion treated *C. albicans* cells had adhered to each other with broken cell walls. On the other hand, Figure 10C outlines the morphological feature of untreated *S. aureus* cells. The cells exhibited an intact round shape with complete cell walls. However, Figure 10D reveals the morphological change of *S. aureus* cells treated with anise oil based-nanoemulsion. The treated cells became deformed and adhered to each other with broken cell walls. The obtained data were in agreement with the fact outlining that some essential oil constituents, including geraniol, menthol, linalool, etc., are terpenoid compounds and their mode of action as bactericidal agents could be attributed to their roles in the disruption of the outer and inner membranes and interaction with membrane protein and intercellular targets [44].

## 4. Conclusions

In this study, different concentrations of anise oil-based nanoemulsions were prepared with a nanoemulsion technique using lecithin as the emulsifying agent with the aid of homogenization and ultra-sonification. The data obtained showed uniform particle sizes and suitable stability, using TEM, DLS, zeta potential, FE-SEM, and optical microscopy to illustrate the successful preparation of anise oil-based nanoemulsion. Thus, it was observed that the utilization of lecithin as a surfactant had a significant effect on the droplet diameter and stability of the prepared nanoemulsions. The anise oil-based nanoemulsion exhibited different biological and biochemical activities, including antimicrobial, antioxidant, and cytotoxic activities. The killing time of these nanoemulsions was also investigated. The mode of action of the nanoemulsions was elucidated by studying the release of reducing sugars and proteins from treated cultures of *S. aureus* and *C. albicans* in addition to conducting FE-SEM studies.

## Figures and Tables

**Figure 1 polymers-13-02009-f001:**
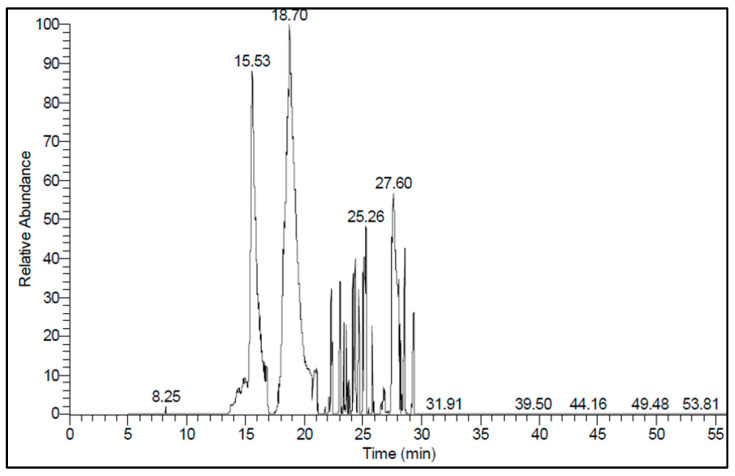
Gas chromatography–mass spectroscopy chromatogram of anise oil.

**Figure 2 polymers-13-02009-f002:**
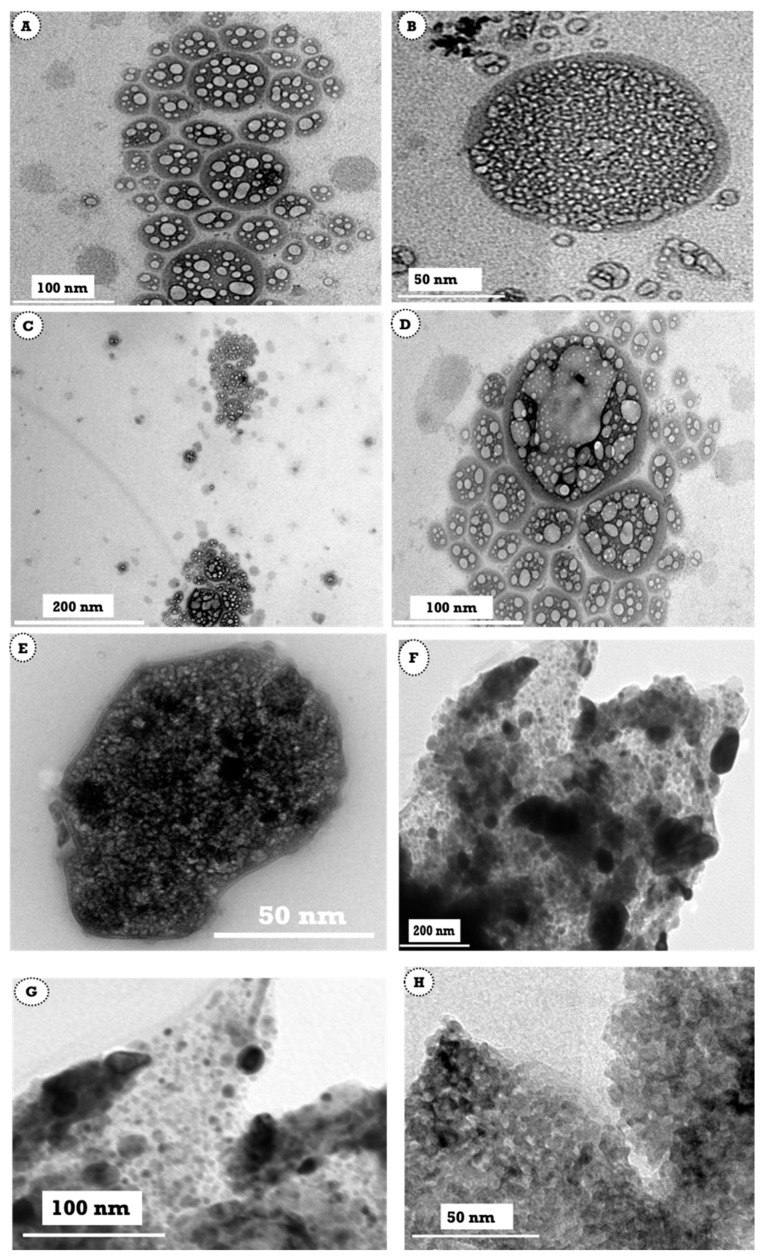
Transmission electron microscopy image of anise oil-based nanoemulsion (**A**,**B**) 5.5 mL, (**C**–**E**) 9.5 mL, and (**F**–**H**) 13.5 mL.

**Figure 3 polymers-13-02009-f003:**
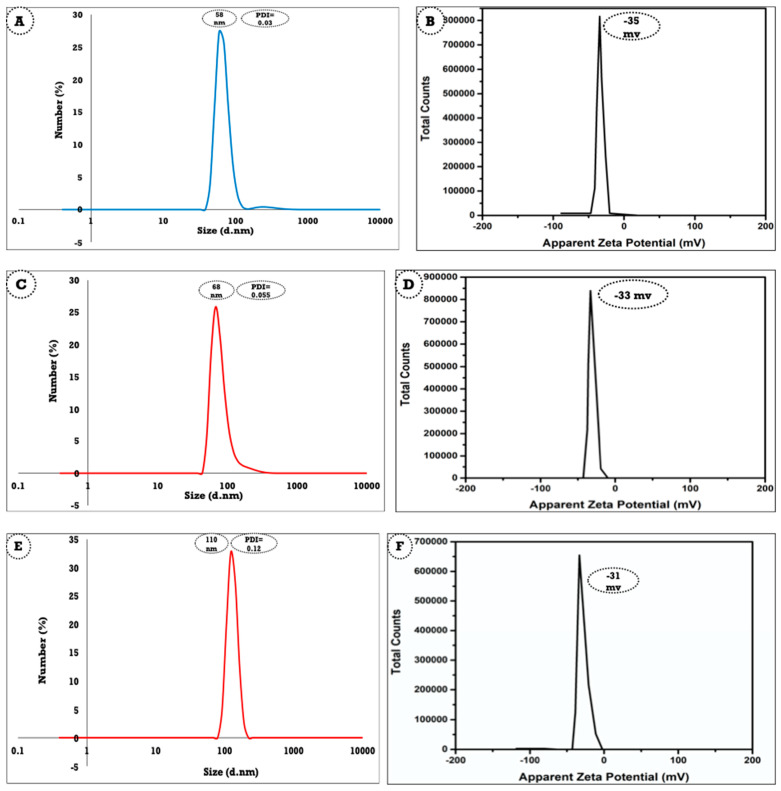
Particle size analyzer and zeta potential of anise oil-based nanoemulsion using (**A**,**B**) 5.5 mL, (**C**,**D**) 9.5 mL, and (**E**,**F**) 13.5 mL of anise oil, respectively.

**Figure 4 polymers-13-02009-f004:**
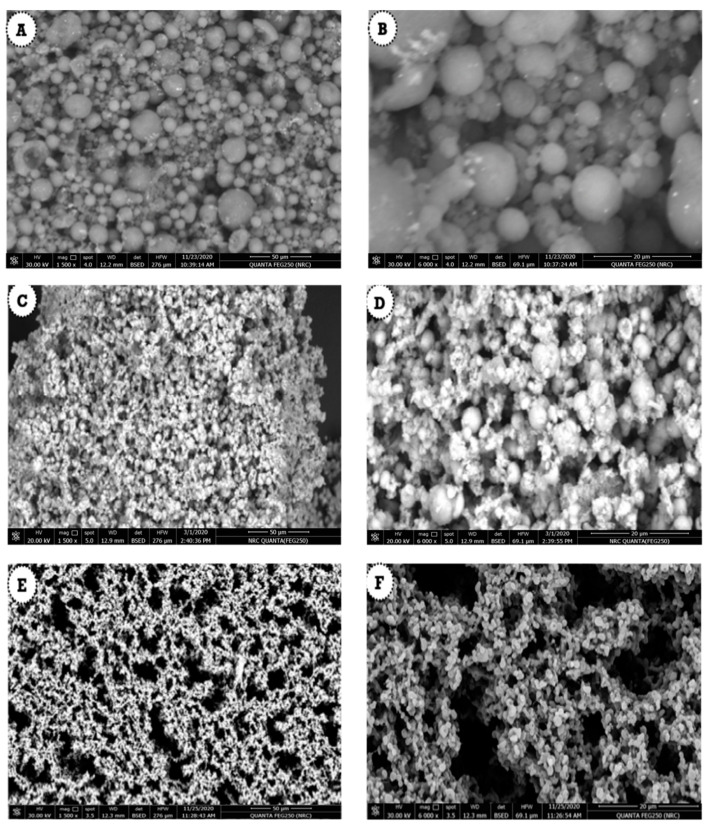
Scanning electron microscope images of anise oil-based nanoemulsions using (**A**,**B**) 5.5 mL, (**C**,**D**) 9.5 mL and (**E**,**F**) 13.5 mL of anise oil.

**Figure 5 polymers-13-02009-f005:**
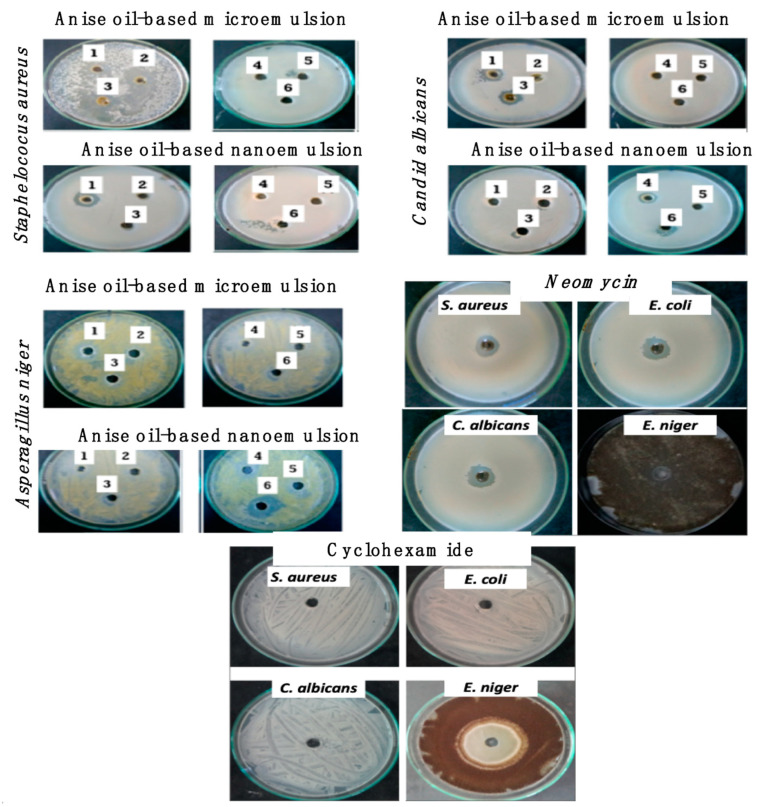
The antimicrobial activities of different concentrations of anise oil-based nanoemulsion against different test microbes as well as antimicrobial references.

**Figure 6 polymers-13-02009-f006:**
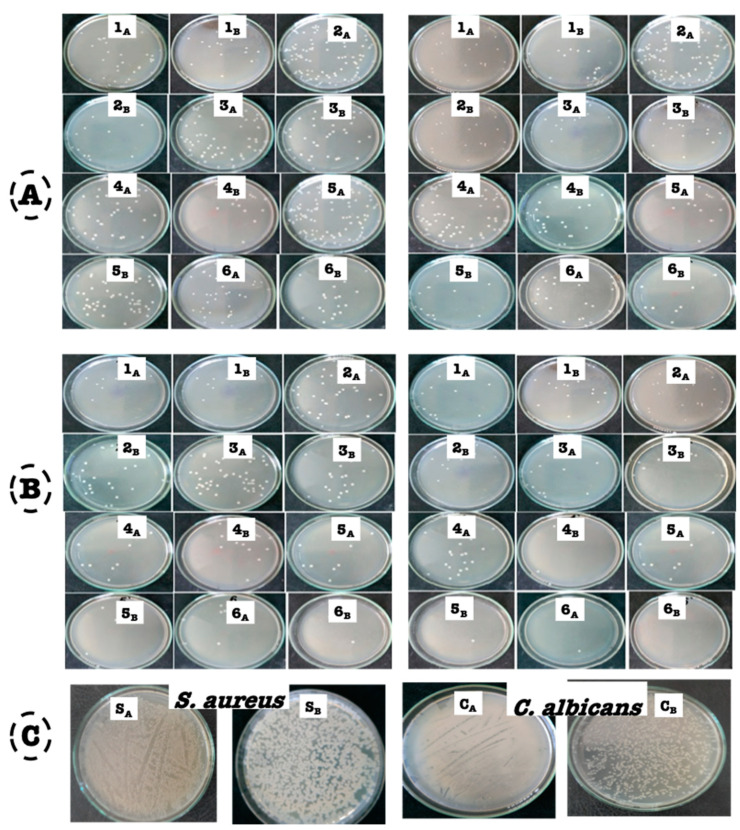
Effect of killing time of anise oil-based nanoemulsion against *C. albicans* and *S. aureus:* (**A**) after 2 h (**B**) after 24 h, _A_-first dilution, _B_-second dilution, (**C**) control-tested microbes; _A, B_, first and second dilutions for *S. aureus* and *C. albicans*, respectively.

**Figure 7 polymers-13-02009-f007:**
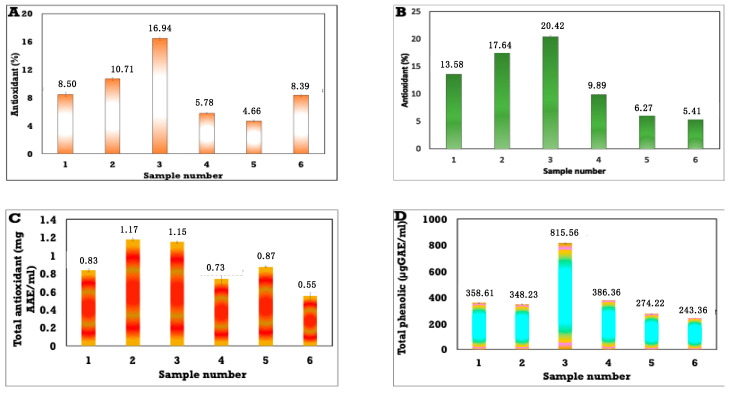
Antioxidant activities of different concentrations of anise oil-based nanoemulsions determined using the 2,2-Diphenyl-1-picrylhydrazyl assay (**A**), 2,2′-Azinobis-3-ethylbenzothiazoline-6-sulfonic acid assay (**B**), total antioxidants (**C**), and total phenolics (**D**).

**Figure 8 polymers-13-02009-f008:**
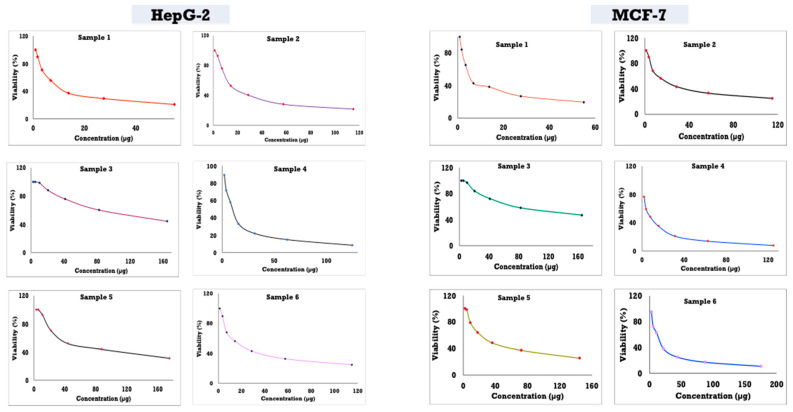
Cytotoxicity of different concentrations of anise oil-based nanoemulsion.

**Figure 9 polymers-13-02009-f009:**
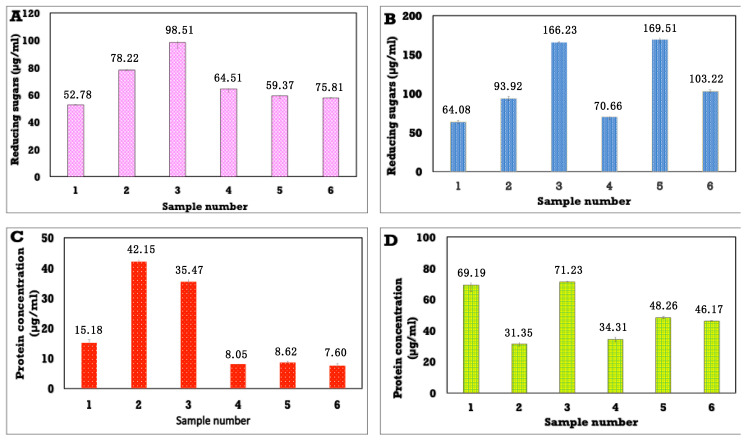
Reducing sugar and protein (µg/mL) released from *S. aureus* and *C. albicans* treated with different concentrations of anise oil-based nanoemulsions. (**A**,**B**): reducing sugar release from *S. aureus* and *C. albicans*, respectively, (**C**,**D**): protein release from *S. aureus* and *C. albicans*, respectively.

**Figure 10 polymers-13-02009-f010:**
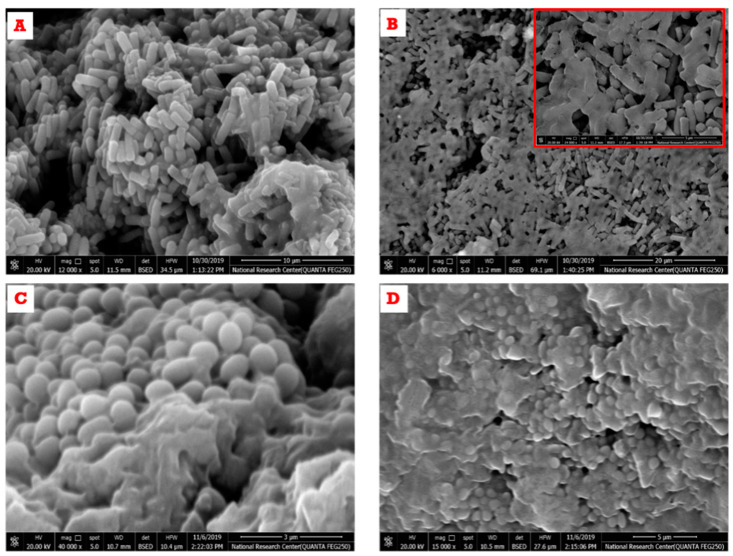
Field emission-scanning electron microscope (FE-SEM) photography of *C. albicans* and *S. aureus* before and after treatment with anise oil-based nanoemulsions. (**A**): untreated *C. albicans* cells, (**B**): treated *C. albicans*, (**C**): untreated *S. aureus* cells and (**D**): treated *S. aureus* cells.

**Table 1 polymers-13-02009-t001:** Chemical constituents of anise oil extracted from the GC/MS analysis.

Sample No.	R_t_	%	M.W.	M.F.	Identified Compounds
1	8.25	0.12	136	C_10_H_16_	Sabinene
2	13.75	0.20	178	C_12_H_18_O	Bicyclo[3.3.1]nonan-2-one, 9-isopropylidene
3	14.22	0.17	154	C_10_H_18_O	Linalool
4	14.27	0.12	136	C_10_H_16_	Camphene
5	14.38	0.21	140	C_9_H_16_O	Nonadien-1-ol
6	14.80	0.22	154	C_10_H_18_O	3-Decen-2-one
7	14.95	0.21	154	C_10_H_18_O	Cyclohexanol,1-methyl-4-(1-methylethenyl)-,cis-6
8	15.53	11.22	154	C_10_H_18_O	1-Menthone
9	15.85	0.17	154	C_9_H_14_O_2_	2-Acetylcycloheptaneone
14	16.12	0.19	154	C_10_H_18_O	3-Hepten-2-one, 3-ethyl-4-methyl
15	16.35	0.16	256	C_14_H_24_O_4_	2,5-Dimethoxy-2,3:5,6-bis(tetramethylene)-1,4-dioxane
16	16.71	0.24	150	C_9_H_10_O_2_	2-Methyl-2-(2′-propynyloxy)-4-pentynal
17	16.84	1.09	154	C_10_H_18_O	Cyclohexanone, 5-methyl-2-(1-methylethyl)-,(2R-cis)-
18	18.13	0.22	156	C_10_H_20_O	α-Citronellol
19	18.24	0.73	250	C_17_H_30_O	2,6-Dimethyl1cyclohexyl-2-ethylenehept-5-enol
20	18.29	0.49	138	C_10_H_18_	1,6-Octadiene,2,7-dimethyl-
21	18.63	4.19	140	C_9_H_16_O	2-Nonyn-1-ol
22	18.71	8.35	124	C_9_H_16_	1,6-Heptadiene,3,3-dimethyl-
23	20.61	1.02	138	C_10_H_18_	1,4-Hexadiene,3-ethyl-4,5-dimethyl
24	21.04	3.48	154	C_10_H_18_O	Geraniol
25	21.13	0.12	182	C_11_H_18_O_2_	Geraniol formate
26	22.11	0.26	204	C_15_H_24_	α-Ylangene
27	22.30	4.03	196	C_12_H_20_O_2_	Geranyl acetate
28	22.38	0.75	204	C_15_H_24_	(-)-α-Bourbonene
29	23.05	4.77	144	C_8_H_16_O_2_	1-Propanol,2-methyl-1-[1-(hydroxymethyl)cyclopropyl]
30	23.38	1.24	204	C_15_H_24_	Cadinene
31	23.58	1.41	204	C_15_H_24_	Calarene
32	23.71	0.37	204	C_15_H_24_	Humulene
33	23.82	0.27	204	C_15_H_24_	Trans-Caryophyllene
34	24.15	3.13	222	C_15_H_26_O	Nerolidol
35	24.35	6.78	204	C_15_H_24_	Gurjunene
36	24.63	3.93	204	C_15_H_24_	Elemene
37	25.01	1.54	220	C_15_H_24_O	Alloaromadendrene oxide (2)
38	25.16	5.19	204	C_13_H_16_O_2_	Cyclopentanecarboxylic acid, 1-(4-methylphenyl)-
39	25.28	6.79	222	C_15_H_26_O	3(R)-Hydroxy-4-acorene
40	25.79	1.30	220	C_15_H_24_O	Cedrene oxide
41	25.90	0.18	224	C_14_H_24_O_2_	Butanoic acid, 3,7-dimethyl-2,6-octadienyl ester, (E)-
42	26.55	0.22	276	C_18_H_28_O_2_	10,12-Octadecadiynoic acid
43	26.74	0.64	204	C_13_H_16_O_2_	2-Phenylethyl tiglate
44	26.85	0.61	204	C_15_H_24_	Trans-Caryophyllene
45	27.49	2.97	222	C_15_H_26_O	gamma-Eudesmol
46	27.59	6.29	204	C_12_H_16_N_2_O	2-Pyrrolidinone,1-(4-amino-3,5-dimethylphenyl)
47	28.08	2.71	326	C_21_H_26_O_3_	Benzoic acid,5-methylene-1,1,4a-trimethyldecalin-6-on-2-ylester (8aá)
48	28.16	0.66	222	C_15_H_26_O	Agarospirol
49	28.22	0.62	222	C_15_H_26_O	Cubebol
50	28.57	6.41	138	C_9_H_14_O	1H-Inden-1-one,octahydro-,cis
51	29.33	3.53	236	C_15_H_24_O_2_	Geranyl tiglate
T%		99.52%			

**Table 2 polymers-13-02009-t002:** The antimicrobial activities of different concentrations of anise oil-based nanoemulsion against different test microbes.

Sample No.	Anise Oil-Based Nanoemulsion Concentration (mg/mL)	Clear Zone (ϕmm)			
		*S. aureus*	*A. niger*	*C. albicans*	*E. coli*
1	0.055	27.5 ± 0.50	0	17.93 ± 0.11	21.96 ± 0.15
2	0.115	22.16 ± 0.28	0	12.46 ± 0.50	20.26 ± 0.25
3	0.165	27.83 ± 0. 76	0	19.26 ± 0.30	24.40 ± 0.40
4	0.096	23.33 ± 0.57	0	20.63 ± 0.55	13.93 ± 0.11
5	0.072	12.40 ± 0.36	0	0	0
6	0.048	13.60 ± 0.52	0	15.03 ± 0.25	20.16 ± 0.29
Neomycin Cyclohexamide	0.1	26.53 ± 0.47	21.86 ± 0.81	28.16 ± 0.28	0
	0.1	0	0	0	31.20 ± 0.20

**Table 3 polymers-13-02009-t003:** MIC and MBC of different concentrations of anise oil-based nanoemulsion against all tested microbes.

Sample No.	Sample conc.(mg /mL)	*A. niger*		*E. coli*		*S. aureus*		*C. albicans*	
		MIC (µg/mL)	MBC (µg/mL)	MIC (µg/mL)	MBC (µg/mL)	MIC (µg/mL)	MBC (µg/mL)	MIC (µg/mL)	MBC (µg/mL)
1	0.055	13.50 ± 0.25	27.16 ± 0.29	27.18 ± 0.27	55.35 ± 0.31	13.76 ± 0.02	27.33 ± 0.28	6.70 ± 0.26	27.80 ± 0.26
2	0.115	14.25 ± 0.25	28.73 ± 0.25	28.70 ± 0.13	57.76 ± 0.25	14.55 ± 0.25	57.56 ± 0.26	7.23 ± 0.15	57.29 ± 0.26
3	0.165	20.25 ± 0.25	41.33 ± 0.14	41.50 ± 0.25	82.85 ± 0.78	20.79 ± 0.19	20.79 ± 0.19	5.25 ± 0.13	41.31 ± 0.16
4	0.096	24.06 ± 0.11	48.33 ± 0.38	24.0 ± 0.25	48.43 ± 0.39	23.58 ± 0.52	24.20 ± 0.18	1.50 ± 0.25	24.0 ± 0.0
5	0.072	9.0 ± 0.20	18.29 ± 0.26	36.05 ± 0.18	72.26 ± 0.25	4.48 ± 0.22	36.20 ± 0.20	18.18 ± 0.16	36.15 ± 0.14
6	0.048	24.08 ± 0.14	24.28 ± 0.30	12.21 ± 0.22	24.23 ± 0.32	3.23 ± 0.21	12.15 ± 0.18	24.11 ± 0.12	12.16 ± 0.15

**Table 4 polymers-13-02009-t004:** Effect of different anise oil-based nanoemulsion concentrations on the mortality of *Staphylococcus aureus* after 2 and 24 h.

Sample No.	Sample Description	Concentration Used(µg/mL)	No. of Colonies (cfu)	
			After 2 h	After 24 h
1	MIC	6.8	37	12
1B	One conc above MIC	13.6	19	11
2	MIC	7.1	41	37
2B	One conc above MIC	14.2	13	30
3	MIC	5.16	40	55
3B	One conc above MIC	10.3	15	21
4	MIC	3	40	8
4B	One conc above MIC	6	18	15
5	MIC	18	57	8
5B	One conc above MIC	36	42	3
6	MIC	24	44	7
6B	One conc above MIC	48	21	2

**Table 5 polymers-13-02009-t005:** Effect of killing time of anise oil-based nanoemulsions against *Staphylococcus aureus* after 2 and 24 h.

Sample No.	Sample Description	Concentration Used(µg/mL)	No. of Colonies (cfu)	
			After 2 h	After 24 h
1	MIC	6.87	25	14
1B	One conc above MIC	13.74	17	18
2	MIC	14.3	96	22
2B	One conc above MIC	28.6	26	14
3	MIC	20.6	17	13
3B	One conc above MIC	41.2	12	12
4	MIC	3	50	20
4B	One conc above MIC	6	25	3
5	MIC	0.56	15	8
5B	One conc above MIC	1.12	12	3
6	MIC	6	28	1
6B	One conc above MIC	12	10	0

**Table 6 polymers-13-02009-t006:** Cytotoxicity activity of anise oil-based nanoemulsions against two cell lines, HepG2 and MCF7.

Sample No.	Concentration of Anise Oil-Based Nanoemulsion (µg/mL)	IC50 (µg)	
		HepG-2	MCF-7
1	55	9.61	7.78
2	115	21.48	23.31
3	165	125.60	127.80
4	96	50.03	43.82
5	72	43.20	23.68
6	48	40.65	31.40

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
