# Peer review of "Facile Synthesis of Natural Anise-Based Nanoemulsions and Their Antimicrobial Activity"

_polymers, 2021, doi:10.3390/polym13122009_

Round 1
Reviewer 1 Report
The paper entitled “Facile synthesis of natural anise-based nanoemulsions and their antimicrobial activity” by El-Naggar et al. investigates the preparation of a series of anise oil –in-water nanoemulsions stabilized with lecithin. Moreover, the biological behavior (antibacterial, cytotoxicity etc) of these nanoemulsions was studied.
The authors have used quite a lot of analysis techniques and the quantity of results is impressive. However, the article has quite a lot of drawbacks:
- An intensive English editing is required.
- The authors use several times the term “nanoaprticles” in order to describe the droplets of the emulsion. I suggest the replacement of this term with “nanodroplets”
- Starting with section 3.5, the authors have used 6 new emulsion samples but without describing the procedure for their preparation. Moreover, the samples should be numbered as a function of the concentration increase, if the authors wanted to study this effect. In the actual stage, it is quite confusing and difficult to make some correlations between the concentration of these samples and the obtained results. Moreover, in table 2 the units are given in mg/ml whereas in table 3 they are given in μg/ml. Why the authors have not just used the first 3 nanoemulsions which were characterized 3.2-3.4?!
- For all the biological tests, the anise oil must be used as reference.
- Line 30: delete “essential oils”; it’s a repetition.
- Lines 79-82: please revise the last sentence from the introduction section
- Line 85: from where the anise oil was purchased?! Please verify
- Line 102: the title of this section should be modified as follows: characterization of anise oil-based nanoemulsions
- Line 131: please revise the sentence: “the samples…in this test.”
- Line 359: MIC and MBC
- Line 360: C. albicans
- Line 391-395: please revise the paragraph.
- Data given in the figures should have at maximum 2 decimals.
- Fig 8. Complete the name of the ordinate with “cells viability”
Author Response
Point-by-point response to reviewers’ comments
Ms. Ref. No.: polymers-1208444
Title: Facile Synthesis of Natural Anise-Based Nanoemulsions and Their Antimicrobial Activity
Correspondence Author: Assist. Prof. Mehrez E. El-Naggar
We are really thankful for giving us the opportunity to revise our manuscript entitled above. I carefully considered the reviewers’ comments. I want to extend my appreciation for taking the time and effort necessary to provide such insightful guidance. The revision, based on the review team’s collective input, includes a number of positive changes. Based on your guidance, I have accordingly modified the manuscript and detailed corrections, changes, and/or rebuttals against each point raised are listed below and addressed in the revised version by track change. I hope that these revisions improve the paper such that you and the reviewers now deem it worthy for publication in the Journal of Polymers.
Here is the point-by-point response to all the comments:
Reviewers' comments:
Reviewer #1:
The paper entitled “Facile synthesis of natural anise-based nanoemulsions and their antimicrobial activity” by El-Naggar et al. investigates the preparation of a series of anise oil –in-water nanoemulsions stabilized with lecithin. Moreover, the biological behavior (antibacterial, cytotoxicity etc) of these nanoemulsions was studied.
The authors have used quite a lot of analysis techniques and the quantity of results is impressive. However, the article has quite a lot of drawbacks:
- An intensive English editing is required.
The language of the whole manuscript has been carefully checked and revised
- The authors use several times the term “nanoaprticles” in order to describe the droplets of the emulsion. I suggest the replacement of this term with “nanodroplets”
The word has been changed to nanodroplets with the manuscript
- Starting with section 3.5, the authors have used 6 new emulsion samples but without describing the procedure for their preparation. Moreover, the samples should be numbered as a function of the concentration increase, if the authors wanted to study this effect. In the actual stage, it is quite confusing and difficult to make some correlations between the concentration of these samples and the obtained results.
Sorry for this confusing, The concentrations have been clarified in the manuscript as follow: Six concentrations of anise oil-based nanoemulsions were selected according to their antimicrobial activities. These six concentration were coded as 1, 2, 3, 4, 5 and 6. These codes samples and their abbrevations were outlined in Table 2. These concentrations were prepared by dilutions of of anise oil-based nanoemuslision that prepared using 9.5 mL of anise oil.
- Moreover, in table 2 the units are given in mg/ml whereas in table 3 they are given in μg/ml. Why the authors have not just used the first 3 nanoemulsions which were characterized 3.2-3.4?!
The unit has been corrected to mg/mL.
- For all the biological tests, the anise oil must be used as reference.
Several previous works have been done on anise oil itself including antimicrobial antioxidant and others and no need to repeat their works in addition the nanoemulsion of the oil is different from the main oil.
- Amer, U. I. Aly Antioxidant and antibacterial properties of anise (Pimpinella anisum L.) Egyptian Pharmaceutical Journal 2019, 18:68–73
Mashareq M. K., **Amira M. E., *** Zenab A.A., *Ali I. A., and * Fathy I. R. EVALUATING ANTIMICROBIAL AND ANTIOXIDANT ACTIVITIES OF VOLATILE OILS EXTRACTED FROM ANISE, FENNEL AND SPEARMINT PLANTS J. Agric. Res. Kafr El-Sheikh Univ. pp: 196-209, Vol. 42(2) 2016
- Line 30: delete “essential oils”; it’s a repetition.
Thanks for your observation. The repeated “essential oils” has been deleted.
- Lines 79-82: please revise the last sentence from the introduction section
The sentence has been checked and amended
- Line 85: from where the anise oil was purchased?! Please verify
Anise oil with a high purity (>99%) was locally obtained from our traditional remarkable market (Egypt).
- Line 102: the title of this section should be modified as follows: characterization of anise oil-based nanoemulsions
The title has been changed as required by Reviewers
- Line 131: please revise the sentence: “the samples…in this test.”
The sentence has been amended to be more readable
- Line 359: MIC and MBC
The abbreviations and their sentences have been amended
- Line 360: C. albicans
The word has been corrected
- Line 391-395: please revise the paragraph.
These two sentences have been checked and amended to be more readable.
- Data given in the figures should have at maximum 2 decimals.
Thanks for your comment, Figure 7 and Figure 9 have been redrawn as recommended by Reviewer.
- Fig 8. Complete the name of the ordinate with “cells viability”
The name of cell viability has been completely written

Reviewer 2 Report
In this paper, different concentrations of anise-based nanoemulsions were prepared using lecithin as the emulsifying agent with the aid of homogenization and ultra-sonification. Different biochemical and biological evaluations of anise oil nanoemulsions were conducted. The paper fit the aims and scope of Polymers. I would recommend accepting the paper after modifications. I have some comments to the authors.
Introduction
This section should be heavily improved. The logic of the present text needs to be improved. For example, line 32-33 seems irrelevant to the context. This paper does not make any exploration on the conditions of ultrasonic processing, and ultrasonic processing is not the focus of this paper. Therefore, it is not recommended to talk too much on the preparation of nanoemulsion by ultrasonic. It should focus on the advantages of emulsification stability and bactericidal activity after the formation of nanoemulsion. The incipit has to be supported with proper suitable literature references. For example: Fabrication and characterization of a microemulsion stabilized by integrated phosvitin and gallic acid. J. Agric. Food Chem. 2020, 68, 5437–5447. doi: 10.1021/acs.jafc.0c00945. Preparation of whey protein isolate nanofibrils by microwave heating and its application as carriers of lipophilic bioactive substances, https://doi.org/10.1016/j.lwt.2020.109213
Line 94-95: Is it molar ratio? Or mass ratio
Line 98: Details of ultrasonic processing should be provided.
Line 157:(ABTS):
Line 192:observe should be changed to evaluate
Section 2.4 is the experiment repeated three times?
Line 215-236: It did not belong results and discussion
Chemical analysis of anise oil shows no practical meaning to this article
Section 3.5 What did “Six concentrations of nanoemulsion” mean? Is the content of anise essential oil in the emulsion different or add different amounts of nanoemulsion to the medium? What is the ratio of the ingredients in the nanoemulsion?
Table 2 and 3 lack of statistical analysis. Are there significant differences in the data?
What were Sample 1 and Sample 1B in Table 4 and 5? And what were sample 2,2B, 3,3B…?Sample 1,2,3…in Figure 7 and 9? Such undefined numbers make reading very difficult.
What does the determination of total phenol content want to show?
Methods should be described in more detail.
Discussion should be greatly improved.
Author Response
Point-by-point response to reviewers’ comments
Ms. Ref. No.: polymers-1208444
Title: Facile Synthesis of Natural Anise-Based Nanoemulsions and Their Antimicrobial Activity
Correspondence Author: Assist. Prof. Mehrez E. El-Naggar
We are really thankful for giving us the opportunity to revise our manuscript entitled above. I carefully considered the reviewers’ comments. I want to extend my appreciation for taking the time and effort necessary to provide such insightful guidance. The revision, based on the review team’s collective input, includes a number of positive changes. Based on your guidance, I have accordingly modified the manuscript and detailed corrections, changes, and/or rebuttals against each point raised are listed below and addressed in the revised version by track change. I hope that these revisions improve the paper such that you and the reviewers now deem it worthy for publication in the Journal of Polymers.
Here is the point-by-point response to all the comments:
Reviewer #2:
In this paper, different concentrations of anise-based nanoemulsions were prepared using lecithin as the emulsifying agent with the aid of homogenization and ultra-sonification. Different biochemical and biological evaluations of anise oil nanoemulsions were conducted. The paper fit the aims and scope of Polymers. I would recommend accepting the paper after modifications. I have some comments to the authors.
Introduction
This section should be heavily improved. The logic of the present text needs to be improved. For example, line 32-33 seems irrelevant to the context.
The current paragraph has been deleted from the manuscript “ the direct incorporation of essential oils into foodstuffs is restricted by numerous of technical challenges related to their lack of diffusion of essential oils in the food matrices, regulating their interaction with other ingredients and the longevity of their activity for the time needed”.
This paper does not make any exploration on the conditions of ultrasonic processing, and ultrasonic processing is not the focus of this paper. Therefore, it is not recommended to talk too much on the preparation of nanoemulsion by ultrasonic.
The paragraph about ultrasonication in the introduction part has been shortened.
It should focus on the advantages of emulsification stability and bactericidal activity after the formation of nanoemulsion. The incipit has to be supported with proper suitable literature references. For example: Fabrication and characterization of a microemulsion stabilized by integrated phosvitin and gallic acid. J. Agric. Food Chem. 2020, 68, 5437–5447. doi: 10.1021/acs.jafc.0c00945. Preparation of whey protein isolate nanofibrils by microwave heating and its application as carriers of lipophilic bioactive substances, https://doi.org/10.1016/j.lwt.2020.109213
The required references have been added to enhance our introduction part.
Line 94-95: Is it molar ratio? Or mass ratio
Thanks for your comment, the addition was carried out using molar ratio.
Line 98: Details of ultrasonic processing should be provided.
The ultrasonication process was carried out at 80 kHz and output powers (200 W) for 30 min at room temperature
Line 157:(ABTS):
Sorry for this mistake, the word has been corrected.
Line 192:observe should be changed to evaluate
The word has been changed from “observe” to “evaluate”.
Section 2.4 is the experiment repeated three times?
Each experiment has been carried out three times.
Line 215-236: It did not belong results and discussion
These paragraphs have been added before discuss the obtained data to summarize the steps of our current work
Chemical analysis of anise oil shows no practical meaning to this article
Thanks for your comment. However, the chemical analysis of anise gives an idea about its constituents which explain its bioactivity. Thus, it is very important firstly to examine the chemical constituents of the pure anise oil.
Section 3.5 What did “Six concentrations of nanoemulsion” mean? Is the content of anise essential oil in the emulsion different or add different amounts of nanoemulsion to the medium? What is the ratio of the ingredients in the nanoemulsion?
The anise oil-based nanoemulsion prepared using 13.5 mL was selected for further characterization . Based on that, 0.055, 0.165, 0.096, 0.072, 0.048 and 0.1 (μg/mL) were produced by sample diluation. These six concentration were coded as 1, 2, 3, 4, 5 and 6. These code samples and their abbrevations were outlined in Table 2. These concentrations were prepared by dilutions of of anise oil-based nanoemuslision that prepared using 9.5 mL of anise oil.
Table 2 and 3 lack of statistical analysis. Are there significant differences in the data?
The values in table 2 and table 4 have been added with statistical analysis.
What were Sample 1 and Sample 1B in Table 4 and 5? And what were sample 2,2B, 3,3B…?Sample 1,2,3…in Figure 7 and 9? Such undefined numbers make reading very difficult.
Six selected concentrations, according to their antimicrobial activities from anise oil-based nanoemulsion. These six concentration were coded as 1, 2, 3, 4, 5 and 6. These code samples and their abbrevations were outlined in Table 2. In Table 4 samples 1, 2, 3, 4, 5 and 6 are the MIC of the used concentrations whereas samples 1B. 2B, 3B, 4B, 5B and 6B are one concentration above the MIC (column 3 from left) (as mentioned in methodology section) for more details we took the MIC for each concentration and one concentration above it and we perform the kill time test
What does the determination of total phenol content want to show?
It gives an idication about the phenolic contents which are mainly responsible for bioactivity
Methods should be described in more detail.
Methods part has been rewritten to be more readable.
Discussion should be greatly improved.
Thanks for your suggestion, discussion part has been rewritten and enhanced to be more readable.

Round 2
Reviewer 1 Report
The paper can be published as it is.
Reviewer 2 Report
Thanks for the corrention.